# Environmental factors influencing the prevention of secondary health conditions among people with spinal cord injury, South Africa

**Sonti Pilusa** [ORCID] *, **Hellen Myezwa**[☉], **Joanne Potterton**[☉]

Faculty of Health Sciences, Department of Physiotherapy, School of Therapeutic Sciences, University of the Witwatersrand, Johannesburg, South Africa

☉ These authors contributed equally to this work.
* Sonti.pilusa@wits.ac.za

**Data Availability Statement:** The transcript data contain potentially identifying information. Deidentified transcripts are available on request. Non-author contact information for an ethics

## Abstract

### Background

The environment where people live, work or play can influence health and disability outcomes. People with spinal cord injury are at risk for secondary health conditions, with this increasing readmission rates and decreasing quality of life. Studies on preventative care for secondary health conditions and factors influencing the prevention of secondary health conditions are scarce in low to middle-income countries.

### Aim

To explore environmental factors influencing the prevention of secondary health conditions in people with spinal cord injury.

### Setting

This study was based at a public rehabilitation hospital, South Africa.

### Methods

Explorative qualitative design was used. Semi-structured interviews were conducted with 21 therapists, 17 people with a spinal cord injury and six caregivers. The interviews were transcribed verbatim. Analysis was conducted using content analysis.

### Results

The categories that emerged included the impact of social support, inaccessible built environment and transport system, and an inefficient health care system. Sub-categories for the inefficient health care systems were: Shortage of resources, health workers lack of knowledge on prevention of secondary health conditions and inadequate patient care approach.

committee to which transcript data requests may be sent is as follows: Human research ethics committee (Medical) University of the Witwatersrand, CONTACT DETAILS Rhulani Mkansi Rhulani.Mkansi@wits.ac.za Ms Zanele Ndlovu Zanele.Ndlovu@wits.ac.za.

**Funding:** SP was supported by the Consortium for Advanced Research Training in Africa (CARTA). CARTA is jointly led by the African Population and Health Research Center and the University of the Witwatersrand and funded by the Carnegie Corporation of New York (Grant No—G-19-57145), Sida (Grant No:54100113), Uppsala Monitoring Centre and the DELTAS Africa Initiative (Grant No: 107768/Z/15/Z). The DELTAS Africa Initiative is an independent funding scheme of the African Academy of Sciences (AAS)'s Alliance for Accelerating Excellence in Science in Africa (AESA) and supported by the New Partnership for Africa's Development Planning and Coordinating Agency (NEPAD Agency) with funding from the Wellcome Trust (UK) and the UK government. The statements made and views expressed are solely the responsibility of the Fellow. Thuthuka grant also supported this research.

**Competing interests:** The authors have declared no competing interest exist.

## Conclusion

Environmental factors influencing the prevention of secondary health conditions are complex and multifactorial. When developing rehabilitation and prevention programmes, environmental factors must be considered.

## Introduction

According to the International Classification of Functioning, Disability and Health (ICF), environmental factors are factors outside an individual (extrinsic) where people live, play or work, which can influence health positively or negatively [1]. Environmental factors include "products and technology, natural environment and human-made changes to the environment, support and relationships, attitudes, services, systems and policies" [2 p16]. The presence of a disability is not always the main problem for people with disabilities, but it is the presence of systematic environmental factors that shape the experience of disability [2]. There is a need to explore environmental factors that influence health outcomes to inform context-based interventions for people with disabilities in South Africa.

A chronic condition, such as spinal cord injury (SCI), requires long-term care, including rehabilitation, to optimize function, health maintenance and disease prevention. Preventative care in people with SCI is neglected even though this population is at a high risk of developing complications that are not necessarily caused by the primary disability but can occur because they have a disability [3]. These complications are called secondary conditions or secondary health conditions (SHCs) [4]. Local studies on SHCs among people with SCI reported a high prevalence of SHCs ranging from 50% during the acute phase [5] to 89% four years post-injury [6]. The most prevalent SHCs were pressure sores, pulmonary complication, and urinary tract infection during the acute phase [5]. Whilst post-discharge, the most prevalent was pain, muscle spasms and sleeping problems [6]. The presence of SHCs dramatically affects the quality of life [7], increases hospital readmission [8] and can lead to untimely death [2]. Although many of these SHCs are preventable, evidence shows that preventing SHCs is challenging [9,10].

There is a dearth of studies on preventive care for SHCs and factors influencing prevention care for people with SCI in low to middle-income countries. Previous studies based in high-income countries (Canada and Switzerland) have reported that prevention of SHCs is influenced by personal prevention style, medically oriented model of care, poor transport and built accessibility, social support systems, health professionals' lack of knowledge on SCI and SHCs, and uncoordinated care [10–12]. Although these studies highlighted relevant factors that influence the prevention of SHCs, the context is not the same as in South Africa. Socioeconomic disparities still exist in South Africa. Many of the population are unemployed and live in poverty (49%) with limited access to social services [13]. This situation is even worse for people with disabilities, with the majority of people with disabilities unemployed and 41% living below the poverty line [14]. Measures to protect vulnerable individuals in South Africa include access to various types of social grants. However, evidence indicates that social grants are not sufficient to meet the needs of an individual with disabilities because of disability-related costs, such as payment for a caregiver, high transport costs and costs related to accessing health services [15].

The South African health care system has a government subsidised private and public health sector, but most of the population (84%) depends on the under-resourced public and government-run health care system [16]. The remaining 16% of the population pay for their

health insurance giving them access to quality medical resources and readily accessible health care services [16]. The inequitable access to health is one reason why the government has initiated the National Health Insurance (NHI) discourse to bridge the gap and ensure every citizen has access to quality and affordable health care [17]. It is important to note that information about rehabilitation services is limited in the NHI, and the implementation of the NHI has not commenced.

Furthermore, the health care system faces a quadruple burden of disease, with most of its resources targeted to managing HIV/AIDS, TB and non-communicable diseases [16]. Care is primarily curative, neglecting long-term impairments and disabilities related to prevalent health problems [18]. As such, rehabilitation care across the continuum is not prioritised, with a shortage of rehabilitation health professionals, limited research in disability needs, poor implementation of disability management frameworks and poor access to community-based rehabilitation [18,19]. Lack of rehabilitation services that include prevention and health promotion care leaves individuals with disabilities unable to function and participate fully in society. The reported prevalence of disability was estimated to be 7.5% in the 2011 census [20], a percentage projected to increase due to the rising aged population and the impairments related to the quadruple burden of disease [18]. This increase in the prevalence of disability will increase the demand for health care, rehabilitation and social welfare. With this in mind, context-based studies are necessary to inform the development of appropriate interventions. Thus, this study explored environmental factors that influence the prevention of SHCs among people with SCI.

## Methods

### Study design

We used a qualitative design to explore the environmental factors influencing the prevention of SHCs in people with SCI. This study is part of a more extensive study that aims to establish factors influencing the prevention of SHCs in people with SCI to inform the development of a prevention model of care [21].

### Setting

Participants were recruited from a government-funded rehabilitation hospital in Gauteng, South Africa, the most populous province. The public rehabilitation hospital caters for people with physical disabilities, including spinal cord injury, from local townships and other provinces due to the lack of rehabilitation hospitals in those provinces. People with SCI may travel up to 100 kilometres to attend their monthly appointments for medication and medical consultation using a hired car because of the lack of wheelchair-accessible public transport.

### Recruitment

The first author, assisted by the employee at the rehabilitation hospital, invited patients with SCI who attended the outpatient clinic. Caregivers of people with SCI who consented to the study were invited to participate. The therapists were recruited in the therapy gym.

### Study population

Therapists were included if they were employed in the rehabilitation hospital and involved in the care of people with SCI. Caregivers of people with SCI, formal or informal, had to be 18 years or older and willing to participate in the study. Individuals diagnosed with SCI, both non-traumatic and traumatic, accessing outpatient medical clinic at the rehabilitation hospital

were invited to participate. Willing participants with SCI 18 years and above were considered regardless of their gender, level and duration of SCI.

## Data collection

An interview guide was developed by the researchers informed by previous studies on SHCs [10] fig. Part A covered the participants' sociodemographic data. Part B included questions and probes on the SHCs commonly experienced by people with SCI, prevention and management strategies used, and factors at a personal and environmental level that influenced the prevention of SHCs. Lastly, questions on barriers and facilitators for the prevention of SHCs were asked. The interview guide was piloted on one therapist in the presence of a researcher experienced in qualitative research to clarify the questions. Semi-structured interviews were conducted by the principal researcher at a venue that was suitable for the participants. Interviews for all the therapists and some of the participants with SCI were conducted at the rehabilitation hospital. All the caregivers were interviewed at home. The average length of the interview was 50 min (range 45 min–1h30 min). Interviews were conducted from July 2018-October 2019 and continued until data saturation was reached when no new information emerged.

## Data analysis

All the interviews were audio-recorded and transcribed verbatim for data analysis. MAXQDA 2018.2 was used to manage and analyse data. The quotes are numbered (SC1 1, Therapist 1, Caregiver 1) to ensure anonymity. Analysis was conducted concurrently using content analysis [22]. The primary investigator read and reread all the transcripts to ensure accurate transcription and get a sense of the content. One interview transcript was coded inductively by the principal investigator and two other researchers independently. After that, a discussion on the broad codes was conducted, and a preliminary coding framework was developed, informed by the aim and the ICF environmental concept. An external researcher experienced in qualitative and public health research coded one transcript, and the categories were compared. The differences in the concepts used in the coding framework were discussed. This process helped to refine the categories. Deductive analysis was conducted on the subsequent transcripts, and similar codes were grouped into sub-categories and categories. A manual display of the categories, sub-categories and codes was conducted by the principal researcher and reviewed by the other researchers. Throughout the research process, the authors held regular debriefing sessions.

## Ethical considerations

The rehabilitation hospital granted permission to use the study site for data collection. All the participants gave written informed consent and permission to record before the interview. This study was approved by the Human Research Ethics Committee of the University (M170938) and registered with the South African National Health Research Database (reference GP201712036).

## Results

### 1.1 Demographic profile

The sample included forty-four participants (21 therapists, six caregivers and 17 participants with SCI). The majority (82%) of the participants with SCI were paraplegic and were unmarried. Four caregivers were female and were formally employed as carers. The therapists

**Table 1. Demographic profile of the participants with SCI.**

| Participants with spinal cord injury (n = 17) | |
|---|---|
| **Age in years** | |
| Mean (SD) | 44.5 (13.1) |
| Range | 27–72 |
| **Gender, n (%)** | |
| Male | 14 (82.4) |
| Female | 3 (17.6) |
| **Employed, n (%)** | |
| Yes | 5 (29.4) |
| No | 12 (70.6) |
| **Education level** | |
| Tertiary education | 4 (23.5) |
| Matric | 5 (29.4) |
| High school | 6 (35.3) |
| Primary School | 2 (11.8) |
| **Time since injury** | |
| Mean (SD) | 9 (7.1) |
| Range (years) | 1–30 |
| **Cause of injury, n (%)** | |
| Trauma | 14 (82.4) |
| Non-trauma | 3 (17.6) |
| **Type of spinal cord injury** | |
| Paraplegia | 14 (82.4) |
| Quadriplegia | 3 (17.6) |
| **Completeness of the injury a** | |
| Incomplete | 4 (23.5) |
| Complete | 13 (76.5) |
| **Level of the injury** | |
| **C1-C4** | 2 (11.8) |
| **C5-T1** | 1 (5.9) |
| **T2-T6** | 3 (17.6) |
| **T7-T12** | 9 (52.9) |
| **L1-L5** | 2 (11.8) |
| **Assistive device** | |
| Wheelchair | 14 (82.3) |
| Walking aid | 2 (11.8)) |
| None | 1 (5.9) |
| **Marital status** | |
| Married/staying with a partner | 7 (41.2) |
| Single | 10 (58.8) |

interviewed represented diverse professions, and the mean working experience was 8.7 years SD (8.5). Tables 1 and 2 outline the demographic profile of the participants.

## 1.2 Qualitative data results

Categories evident in the qualitative analysis included the impact of social support, inaccessible built structures and transport system, and an inefficient health care system.

**Table 2. Demographic data for the caregivers and the therapists.**

| Caregivers (n = 6) | |
|---|---|
| **Age in years** | |
| Mean (SD) | 53.9 (12.9) |
| Range | 33–69 |
| **Gender, n (%)** | |
| Female | 4 (66.7) |
| Male | 2 (33.3) |
| **Employed, n (%)** | |
| Yes | 5 (83.3) |
| No | 1 (16.7) |
| **Caregiver role, n (%)** | |
| Formal (i.e. unemployed) | 4 (66.7) |
| Informal (i.e. Family member) | 2 (33.3) |
| **Education, n (%)** | |
| No schooling | 2 (33.3) |
| Primary school | 2 (33.3) |
| High school | 2 (33.3) |
| Therapists (n = 21) | |
| **Age in years** | |
| Mean (SD) | 31.5 (8.3) |
| Range | 22–54 |
| **Gender, n (%)** | |
| Female | 17 (81) |
| Male | 5 (19) |
| **Professions, n (%)** | |
| Occupational therapy | 7 (33.3) |
| Physiotherapy | 6 (28.6) |
| Social worker | 1 (4.8) |
| Psychologist | 1 (4.8) |
| Speech therapist | 2 (9.5) |
| Dietician | 3 (14.3) |
| Occupational therapist assistant | 1 (4.8) |
| **Work experience in years** | |
| Mean (SD) | 8.7 (8.5) |
| Range | 1–28 |

**1.2.1 Impact of social support.** Social support from family, caregiver and peers aided the prevention of SHCs. Social support manifested in home adaptation, emotional encouragement, peers sharing experiences, physically assisting with self-care and financial support.

"*Patients with a good family support do very well. . ., they do not come back with secondary complications*"

(Therapist 6)

"*My support system has collapsed . . . That is why I ended up developing bedsores*"

(SCI 3)

The participants highlighted the importance of the family's knowledge levels on SCI and SHCs in preventing SHCS. If the family member or caregiver lacks knowledge on SHCs, they will not support prevention care.

"*When my mother was bathing me. . .she saw a bedsore and was not aware of what it is*"

(SCI 5)

"*If the caregiver does not know what to do and the patient cannot self-manage . . . t is trial and error then it is a problem*"

(Therapist 19)

**1.2.2 Inaccessible built structures and transport system.** Inaccessible built structures were a barrier to the prevention of SHCs. Built structures in public facilities such as rough terrain, public toilets not suitable for wheelchair users, and uneven sidewalks proved to be barriers.

"*some when they move from here they end up being in the rural area, and they cannot even get out of the gate because of the roads, they end up being depressed in the house*"

(Therapist 2)

"*I came from the hospital on the sidewalk. It is unreal how many times I had to get off the pavement and back on because there is some or other obstacle on the pavement or the cement is uneven*"

(SCI 16)

The participants mentioned how the public transport system was a barrier to the prevention of SHCs. Stigma against and segregation of people with SCI often characterise the public transport system. Participants reported how taxi drivers did not accommodate people using a wheelchair because it takes time to transfer them into the vehicle. Another stressor related to transport was the high cost of hiring an alternative private transport system.

"*Sometimes when you have to come for a check-up and getting a taxi is challenging, you have to hire a private car, and this is expensive*"

(SCI 1)

"*90% of our patients use public transport like the taxi . . ..taxi drivers either do not want to stop for them to help them in or when they do go on the taxi they have to pay for two seats because they have a wheelchair*"

(Therapist 8)

**1.2.3 Inefficient health care system.** Participants reported how inefficiencies in the health care system hampered the prevention of SHCs. The sub-categories included a shortage of resources and a lack of knowledge on SHCs among health professionals.

*1.2.3.1 Shortage of resources.* Participants reported a shortage of medication, consumables for bowel and bladder management and assistive devices. Patients expressed that they run out of medication because they are usually not given enough to last them till the next check-up. "*I ask them for tramadol (pain medication), and they give you one box, and it is meant to last you for three months. . . sometimes they run out, and you have to wait for that three months*" (SCI 5).

The therapist reported that shortage of medication was at times due to budget constraints "*We have shortages of Gabapentin medication because of budget constraints*" (Therapist 16).

Participants with SCI reported that consumables for bowel and bladder management were not always available and were costly if bought privately "*Coloplast I get from the rehabilitation hospital. One kit I get every six months they supply to me. Unfortunately, they do not always have it available, and if you have to go and buy it, I think it's about R1,500 for that which is very, very expensive*" (SCI 15). Unfortunately, lack of necessary bowel and bladder management consumables increased the risk for infection because individuals with SCI ended up reusing consumables. "*Sometimes we run out of stock you may find that a person is using one catheter for almost six weeks*" (Therapist 3).

Access to assistive devices such as wheelchairs, wheelchair cushions and special aerated mattresses helped reduce the risk for pressure sores as expressed by this participant with SCI "*...the major thing I also feel for me not getting pressure sores being in a chair is the cushion I'm sitting on is one of those air cushions, the Roho cushions, 'cos you do not need to do that constant pressure release......I took out the money I saved up to buy myself this cushion*" (SCI 15).

Some participants ended up buying assistive devices privately because of the poor quality assistive devices issued at the hospital "*The wheelchair tyres the rehabilitation hospital gives you is ten times heavier than the ones I have got now (privately)*" (SCI 17).

*1.2.3.2 Lack of knowledge on SHCs among health professionals.* Participants with SCI also expressed how health care workers, including community-based health workers, lacked knowledge on SHCs and prevention.

"*I was attended to by home-based care nurses....they do not know much about bedsore*"

(SCI 3)

"*Even at the local clinic . . . They are aware that there is something called pressure sores, but they do not know how to treat it*"

(SCI 2)

Where there were hospital standard protocols to prevent pressure sores, health professionals did not comply.

"*Then they (nurses) told me I have to turn after every 3 hours, but sometimes it happened that 5 hours passed without nurses turning me, so I think that is the cause of the bedsores*"

(SCI 2)

*1.2.3.3 Inadequate patient care approach.* The participants with SCI described how patient care was not holistic, "*Sometimes the wheelchair would give me problems, I will come and explain that I have a problem with the wheelchair they would fix it, but they would not see that the wheelchair cushion needs to be changed*" (SCI 5). Also, care during the in-patient rehabilitation phase was not empowering patients to self-manage post-discharge as described by these therapists:

"*We rehab our patient to function in a hospital setting that is one of my biggest problem*" (Therapist 2) "*We mainly focus on secondary complications when it comes closer to the discharge time*"

(Therapist 20).

## Discussion

The study aimed to explore environmental factors influencing the prevention of SHCs in people with spinal cord injury by interviewing people with SCI, caregivers and therapists. The environmental factors identified were: the impact of social support, the inaccessibility of built structures and the transport system, and an inefficient health care system.

In agreement with previous studies, good social support from a caregiver and peers enhanced the prevention of SHCs [23,24]. Caregivers are part of a trusted social network helping in the prevention and management of SHCs, offering emotional support, assisting with activities of daily living and linking people with SCI with the health care system [23]. Given the critical role caregivers play, the lack of knowledge on SCI and SHCs is worrying because they will not support prevention care. Peer involvement helps in sharing experiences, expectations, learning and mentoring [25]. Thus, it can be an adjunct to rehabilitation programmes [26]. Given the shortage of health professionals in South Africa, involving the family/caregivers and peers in long-term rehabilitation and preventive care can help yield better health outcomes. Future research in South Africa can explore the role of caregivers and peer support in delivering rehabilitation service delivery.

Despite the United Convention on the Rights of Persons with Disabilities (UNCRPD) directive on universal access [27] and the Sustainable Development Goal 11 on the inclusive and accessible environment [28], environmental barriers for people with disabilities persist. In this study, inaccessible public transport systems and built structures were barriers to the prevention of SHCs. The main environmental barriers include inaccessible transport and buildings and a negative attitude towards people with disabilities from the community, friends and family [15,29]. Given the social inequities in South Africa, the inaccessible infrastructure and transport system limits participation in the labour market, access to leisure and health-promoting care services and perpetuates exclusion and poverty [29–31]. Possible reasons for barriers in accessing transport and built structures is poor implementation and enforcement of the UNCRPD convention article 9 [18,32]. For the private taxi and bus industry, lack of awareness on universal design and government support to make the vehicles accessible for people with disabilities could be reasons for non-compliance. There is a need to promote universal design as a public good. Strategies that can be explored include awareness campaigns on universal design at all levels of service delivery for both public and private sectors. Secondly, lobbying relevant departments to implement and enforce building regulations for all new infrastructure development and changing public transport vehicle specifications to be accessible for wheelchair users.

Similar to previous studies, the health system influenced the prevention of SHCs [10,11]. Shortage of medical resources, health professionals lack of knowledge on prevention of SHCs, and a patient care approach that is not holistic and empowering affected the prevention of SHCs. Comparing the study results with previous studies based in high-income countries, countries with better resources and a NHI, participants in this study and the previous ones highlighted how health professionals lacked information on the prevention of SHCs [11,33]. This finding indicates gaps in the training of health professionals on disability issues, comprehensive chronic care management and general public health promotion. What was also worrying was that patient care protocols that required prevention care practice were not adhered to, proving that there is low value placed on preventative and rehabilitation care. Then again, non-compliance to prevention protocols could be due to the shortage of health professionals to ensure adequate patient care.

The SCI care model seems to be more medically oriented. In this study, patient care was not holistic and not empowering persons with SCI to self-manage. Self-management practice

is key to owning personal health and practising health maintenance [24]. The possible reasons for not empowering people with SCI could be poor integration of public health in health care service delivery, not valuing rehabilitation care in the same manner as curative care, and poor understanding of rehabilitation [18]. According to the WHO, rehabilitation is a continuous process of enabling an individual with a chronic condition and disability to function and participate in society through therapy, health promotion and disease prevention [34]. In light of the burden of chronic diseases facing South Africa, there is a need to prioritise rehabilitation care at all levels of care, including health maintenance and prevention of diseases. Prioritising rehabilitation care can influence planning and resources allocation needed to enhance health for people with disabilities.

The shortage of resources such as medicines, bowel and bladder consumables, and assistive devices affects the prevention of SHCs. The shortage of medication is a national problem due to inadequate procurement processes and suppliers [35]. Also, not prioritising rehabilitation care affects planning and resource allocation for medication and assistive devices needed by people with disabilities [19,29]. Shortage of assistive devices is a common problem in low and middle-income countries [36,37]. Cited reasons for the shortage of assistive devices include lack of budget allocation, inefficient procurement processes, lack of maintenance and repair services, and supplier backlogs [37]. Lack of essential medical resources drives patients to buy privately, increasing out-of-pocket costs and increasing financial vulnerability.

Access to health for people with disabilities, including SCI, is a human right that must be promoted. Since South Africa has ratified the CRPD, issues experienced by people with SCI can be addressed through UN reporting processes. Firstly, the government can allocate the role of monitoring the convention's implementation to a specific department (e.g. Department of Planning, Monitoring and Evaluation). This department can collaborate at different levels with different stakeholders (respective government departments, NGOs for people with disabilities, private sector and training institutions) to monitor implementation and identify implementation problems. A coordinated and integrated approach to implementing the convention can influence planning, resource allocation, and service delivery to promote people with disabilities in all spheres.

This study has some limitations. We only looked at environmental factors for the participants at one rehabilitation hospital. Thus the findings cannot be generalised to other settings. The sample was purposive, which could imply responder bias. The majority of the participants with SCI were male and had traumatic SCI, consistent with local studies on SCI. Future research should include more females and people with non-traumatic SCI. We had very few participants with a higher level of injury. People with higher-level lesions experience more impairments and functional limitations, increasing the risk for SHCs development and hospital readmission [8]. Also, people with high lesions could experience more significant barriers returning to the hospital for follow-up, such as inaccessible public transport, high cost to hire private transport, need for caregiver support, and being less likely to be recruited. Future studies are needed to explore specific factors related to people with higher-level lesions.

## Conclusion

Previous studies have highlighted the need to strengthen the prevention of SHCs among people with SCI [8,26]. One way of strengthening preventative care is to understand the contextual influencing factors. The environmental factors influencing the prevention of SHCs are complex and multiple. There are more factors in the healthcare system that influence the prevention of SHCs as stated by Guilcher et al. [10]. Understanding local problems and context-specific factors in terms of resources, accessibility, health professional competence and patient

care is vital given the high burden of diseases and massive inequality and poverty in South Africa. Secondly, these factors highlight the gaps in promoting human rights for people with disabilities.

## Implications

### Clinical practice

Understanding the complexity of the factors that act as barriers or facilitators to preventing secondary health conditions is vital when planning rehabilitation care services. When developing rehabilitation and prevention programmes, environmental factors must be considered.

### Educational

The results indicated a lack of awareness of SHCs among community-based health professionals. Training and continuing professional development for health professionals can include disability management and prevention of SHCs.

### Research

This study only focused on environmental factors. Future research could explore personal factors influencing the prevention of SHCs in people with SCI.

## Acknowledgments

I thank all the individuals who participated in this study.

## Author Contributions

**Conceptualization:** Sonti Pilusa, Hellen Myezwa, Joanne Potterton.

**Data curation:** Sonti Pilusa.

**Formal analysis:** Sonti Pilusa.

**Funding acquisition:** Sonti Pilusa.

**Investigation:** Sonti Pilusa.

**Methodology:** Sonti Pilusa, Hellen Myezwa, Joanne Potterton.

**Project administration:** Sonti Pilusa.

**Resources:** Sonti Pilusa.

**Software:** Sonti Pilusa.

**Supervision:** Hellen Myezwa, Joanne Potterton.

**Validation:** Sonti Pilusa.

**Visualization:** Sonti Pilusa.

**Writing – original draft:** Sonti Pilusa.

**Writing – review & editing:** Sonti Pilusa, Hellen Myezwa, Joanne Potterton.

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
