## [Decision Letter · Decision Letter 0]

20 Oct 2020

PONE-D-20-20175

“My support system has collapsed, that’s why I ended up developing bedsores”: environmental factors and secondary health conditions among people with spinal cord injury.

PLOS ONE

Dear Dr. Pilusa,

Thank you for submitting your manuscript to PLOS ONE. After careful consideration, we feel that it has merit but does not fully meet PLOS ONE’s publication criteria as it currently stands. Therefore, we invite you to submit a revised version of the manuscript that addresses the points raised during the review process.

We look forward to receiving your revised manuscript.

Kind regards,

Juliet Kiguli, MA, PhD

Academic Editor

PLOS ONE

Journal Requirements:

2. Please include additional information regarding the interview guide used in the study and ensure that you have provided sufficient details that others could replicate the analyses.

For instance, if you developed a guide as part of this study and it is not under a copyright more restrictive than CC-BY, please include a copy, in both the original language and English, as Supporting Information.

In addition, please include any further details about the development and validation of this tool.

4. Please amend the manuscript submission data (via Edit Submission) to include author Hellen Myezwa.

5. Please amend your list of authors on the manuscript to ensure that each author is linked to an affiliation. Authors’ affiliations should reflect the institution where the work was done (if authors moved subsequently, you can also list the new affiliation stating “current affiliation:….” as necessary).

6. Please include a separate caption for each figure in your manuscript.

Additional Editor Comments:

Thank you for your article. Please kindly address the reviewer's comments and also proof read the word to correct any English errors.

Reviewers' comments:

Reviewer's Responses to Questions

**Comments to the Author**

1. Is the manuscript technically sound, and do the data support the conclusions?

Reviewer #1: Partly

2. Has the statistical analysis been performed appropriately and rigorously? 

Reviewer #1: N/A

3. Have the authors made all data underlying the findings in their manuscript fully available?

Reviewer #1: Yes

4. Is the manuscript presented in an intelligible fashion and written in standard English?

Reviewer #1: Yes

5. Review Comments to the Author

Reviewer #1: Thank you for the opportunity to review your manuscript. Your title is engaging and your work investigates an important topic in SCI management.

General comments:

• When referring to your participants with SCI, please refer to them as “participants”, rather than “patients.” While it is appropriate to use the term “patient” when individuals are in-hospital, or receiving direct medical or rehabilitative care, people with SCI who have enrolled in your study and who are presumably individuals in the community should be referred to as participants to reflect participatory nature of their study participation. Please refer to: Harvey, Lisa A. 2019. “Words Matter. Spinal Cord Asks Authors to Choose Their Words Carefully.” Spinal Cord 57(4):257–257.

• Your manuscript states that studies of environmental impacts on SHC prevention are limited in low-middle income countries. There is quite a lot in the literature for higher income countries, and persistent barriers arise regardless of the type of health care system, but for different reasons. That said, what are the underlying causes of your findings that are specific to how rehabilitation / health care is delivered in South Africa? As this is an international journal, your readership may not have a base understanding of your system. Reading your manuscript, I wanted to have a better understanding of the differences between your healthcare system and the one in my country. Please include a concise but informative description of how it works in your country (Private system? Public system? Combination of both? Who typically pays for what and when?). I encourage you to think broadly about the entire welfare regime – in other words – how your nation protects its most vulnerable not only in terms of health care, but also social resources for community living. Then, in the discussion, you can speculate how your results are tied to the structural barriers imposed by the health and welfare system, as appropriate.

• You introduce new results / material in the discussion and the conclusion (a figure). Please restructure appropriately.

Introduction:

• Lines 64-65, surprising that respiratory complications are missing from list of secondary health conditions, unless that is less of a problem in South Africa than in other regions.

• You discuss the Guilcher study, and then allude to differences between Canadian and South African health care systems. As I indicated under general comments, a better understanding of the South African system would strengthen your manuscript. While you indicate that South Africa does not have national health insurance, you indicate that 84% are reliant on the public sector, presumably supported by the government. Please clarify how it generally works, including information about the other 16% - are they wealthy and pay privately?

• There are persistent links between disability and poverty world-wide, in low income to high income countries. You indicate that a large proportion of the general population in South Africa lives in poverty. Can you comment on the proportion of people with disabilities living in poverty in South Africa?

Methods:

• Please provide more detail about recruitment, were prospective participants personally invited (if so, by whom), were there flyers, did you go to the therapy gym to recruit therapist participants, did SCI participants identify the caregiver participants, etc.

• Describe purposive recruitment relative to your recruitment goals. Did you seek to recruit a wide range of injury levels, etc.? What other characteristics were considered?

• Is it possible to include your interview guide as an appendix, or provide examples of questions / topics that were addressed?

• Line 109: Did all interviews last 60 minutes? In results, may want to indicate average length and range.

• Did you begin coding with a preliminary set of codes from your interview guide? You discussed the ICF in the introduction. Did you use the ICF as a theoretical framework and if so, how did it influence your analysis?

Results:

• You have a very low number of participants with cervical level SCI. Please provide (in the discussion as relevant) reasoning for this – I wonder if people with higher level injury experience greater barriers returning to the hospital for follow-up and were thus less likely to be recruited.

• Line 167: Please de-identify the name of the hospital / rehab center.

• Your section on Health care system inefficiencies would benefit, as previously suggested, from a brief but informative overview of how health care and rehabilitation are managed in South Africa.

• The latter half of the results section gets a bit thin, with several headings without substantive content or thick description.

• In general, I think your results are important, but not surprising to people who work in SCI. Most if not all of the issues you report are experienced by people with SCI, to some extent, even in wealthier countries, but for different reasons. Since you acknowledge that there is a dearth of information in low-middle income countries about environmental factors that influence SHC prevention, you could spend some time in the discussion addressing why the results are the same or different (more below).

Discussion:

• Your discussion section should lead off with your most important finding(s), then include additional literature as relevant. As currently written, you begin reviewing literature without context to your findings. Indicate how your findings are congruent or incongruent with the studies you reference in the first paragraph.

• Lines 261-263 – this is a result, and should be reported under results in order to be addressed in the discussion

• It is great that you discuss CRPD and SDGs. Since South Africa has ratified the CRPD, issues experienced by people with SCI can be addressed through UN reporting processes – please discuss how this has the potential to improve rights of people with disabilities.

• Since the goal of your manuscript is to investigate SHC prevention in a lower-income region, it seems important for you to address / speculate about the root causes of your findings. What contributes to the shortage of resources (meds, supplies, assistive technology, etc.)? Lack of government investment? Again, it will be helpful for the reader to have a basic understanding of the South African health care system. For example, in the United States, access barriers can arise from insurers’ unwillingness to pay for certain resources, and if a person cannot pay privately, they do without. Are the shortages in South Africa different in that resources are unavailable even if someone was wealthy enough to pay privately?

• Similarly, regarding prevention and care – what contributes to your reported non-compliance of health workers to various protocols – are they onerous to follow, lack of knowledge about their existence? What are the institutional influences that contribute?

• In general, you could shorten the discussion somewhat by concisely addressing the issues specific to your findings and your overall goal of examining environmental barriers to SHC prevention in a lower-income nation.

Conclusion

• Conclusion: You introduce new material in the conclusion – Figure 1. This material would be better placed in the discussion, or perhaps even results section.

References:

• Please correct reference #9

6. PLOS authors have the option to publish the peer review history of their article (what does this mean?). If published, this will include your full peer review and any attached files.

Reviewer #1: No

---

## [Author Response · Author response to Decision Letter 0]

15 Jan 2021

Table 1: Editors comments 

Corrections - highlighted in grey

Editors Comments corrections Section 

1. Please ensure that your manuscript meets PLOS ONE's style requirements, including those for file naming. Revised 

Please include additional information regarding the interview guide used in the study and ensure that you have provided sufficient details that others could replicate the analyses.

For instance, if you developed a guide as part of this study and it is not under a copyright more restrictive than CC-BY, please include a copy, in both the original language and English, as Supporting Information.

In addition, please include any further details about the development and validation of this tool. Details on the interview guide included in the data collection section 

Supplementary document added 

Details on the interview guide included in the data collection section Line 126-131

We note that you have indicated that data from this study are available upon request. PLOS only allows data to be available upon request if there are legal or ethical restrictions on sharing data publicly. For information on unacceptable data access restrictions, please see http://journals.plos.org/plosone/s/data-availability#loc-unacceptable-data-access-restrictions.

If there are no restrictions, please upload the minimal anonymized data set necessary to replicate your study findings as either Supporting Information files or to a stable, public repository and provide us with the relevant URLs, DOIs, or accession numbers. Please see http://www.bmj.com/content/340/bmj.c181.long for guidelines on how to de-identify and prepare clinical data for publication. For a list of acceptable repositories, please see http://journals.plos.org/plosone/s/data-availability#loc-recommended-repositories

Added file as supporting information 

Please amend the manuscript submission data (via Edit Submission) to include author Hellen Myezwa.

 Amended 

Please amend your list of authors on the manuscript to ensure that each author is linked to an affiliation. Authors’ affiliations should reflect the institution where the work was done (if authors moved subsequently, you can also list the new affiliation stating “current affiliation:….” as necessary). Corrected 

Please include a separate caption for each figure in your manuscript.

 Figure removed 

Please kindly address the reviewer's comments and also proof read the word to correct any English errors. Revised 

Table 2: Reviewers comments 

Reviewers comments Corrections Section

General comments

 When referring to your participants with SCI, please refer to them as “participants”, rather than “patients.” While it is appropriate to use the term “patient” when individuals are in-hospital, or receiving direct medical or rehabilitative care, people with SCI who have enrolled in your study and who are presumably individuals in the community should be referred to as participants to reflect participatory nature of their study participation. Please refer to: Harvey, Lisa A. 2019. “Words Matter. Spinal Cord Asks Authors to Choose Their Words Carefully.” Spinal Cord 57(4):257–257.

 Corrected throughout the manuscript 

Patients with SCI replaced with participants or people with SCI 

Your manuscript states that studies of environmental impacts on SHC prevention are limited in low-middle income countries. There is quite a lot in the literature for higher income countries, and persistent barriers arise regardless of the type of health care system, but for different reasons. That said, what are the underlying causes of your findings that are specific to how rehabilitation / health care is delivered in South Africa? As this is an international journal, your readership may not have a base understanding of your system. Reading your manuscript, I wanted to have a better understanding of the differences between your healthcare system and the one in my country. Please include a concise but informative description of how it works in your country (Private system? Public system? Combination of both? Who typically pays for what and when?). I encourage you to think broadly about the entire welfare regime – in other words – how your nation protects its most vulnerable not only in terms of health care, but also social resources for community living. Then, in the discussion, you can speculate how your results are tied to the structural barriers imposed by the health and welfare system, as appropriate. Revised the introduction and added a section on the South African context and the health care system Line 67-101

• You introduce new results / material in the discussion and the conclusion (a figure). Please restructure appropriately. Removed the figure 

Introduction

• Lines 64-65, surprising that respiratory complications are missing from list of secondary health conditions, unless that is less of a problem in South Africa than in other regions.

 Revised Line 60-65

You discuss the Guilcher study, and then allude to differences between Canadian and South African health care systems. As I indicated under general comments, a better understanding of the South African system would strengthen your manuscript. While you indicate that South Africa does not have national health insurance, you indicate that 84% are reliant on the public sector, presumably supported by the government. Please clarify how it generally works, including information about the other 16% - are they wealthy and pay privately?

• There are persistent links between disability and poverty world-wide, in low income to high income countries. You indicate that a large proportion of the general population in South Africa lives in poverty. Can you comment on the proportion of people with disabilities living in poverty in South Africa? Revised the introduction and added a section on the South African context and the health care system Line 67-101

Methods

Please provide more detail about recruitment, were prospective participants personally invited (if so, by whom), were there flyers, did you go to the therapy gym to recruit therapist participants, did SCI participants identify the caregiver participants, etc.

 Added a section on recruitment Line 126-129

Describe purposive recruitment relative to your recruitment goals. Did you seek to recruit a wide range of injury levels, etc.? What other characteristics were considered? Added in Line 116-122

Is it possible to include your interview guide as an appendix, or provide examples of questions / topics that were addressed?

 Included topics covered and added the interview guide as a supplementary document Line 126-129

Line 109: Did all interviews last 60 minutes? In results, may want to indicate average length and range. Revised the sentence Line 134

 Did you begin coding with a preliminary set of codes from your interview guide? You discussed the ICF in the introduction. Did you use the ICF as a theoretical framework and if so, how did it influence your analysis?

 Revised 140-144

Results

You have a very low number of participants with cervical level SCI. Please provide (in the discussion as relevant) reasoning for this – I wonder if people with higher level injury experience greater barriers returning to the hospital for follow-up and were thus less likely to be recruited. Added a possible reason for low number of participants with cervical level SCI in the paragraph on the study limitations Line 339-348

Line 167: Please de-identify the name of the hospital / rehab center. Revised –removed the name in the quote Line 191

Your section on Health care system inefficiencies would benefit, as previously suggested, from a brief but informative overview of how health care and rehabilitation are managed in South Africa. Overview of the health system outlined in the introduction 

The latter half of the results section gets a bit thin, with several headings without substantive content or thick description. Revised Line 203-250

Discussion

In general, I think your results are important, but not surprising to people who work in SCI. Most if not all of the issues you report are experienced by people with SCI, to some extent, even in wealthier countries, but for different reasons. Since you acknowledge that there is a dearth of information in low-middle income countries about environmental factors that influence SHC prevention, you could spend some time in the discussion addressing why the results are the same or different (more below).

 Revised the whole discussion section 

Your discussion section should lead off with your most important finding(s), then include additional literature as relevant. As currently written, you begin reviewing literature without context to your findings. Indicate how your findings are congruent or incongruent with the studies you reference in the first paragraph Revised Line 256-258

Lines 261-263 – this is a result, and should be reported under results in order to be addressed in the discussion Revised 

 Line 263-265

It is great that you discuss CRPD and SDGs. Since South Africa has ratified the CRPD, issues experienced by people with SCI can be addressed through UN reporting processes – please discuss how this has the potential to improve rights of people with disabilities. Added a section on how the issues experienced by people with SCI can be addressed Line 329-337

Since the goal of your manuscript is to investigate SHC prevention in a lower-income region, it seems important for you to address / speculate about the root causes of your findings. Added the reasons for the findings in the discussion section 

What contributes to the shortage of resources (meds, supplies, assistive technology, etc.)? Lack of government investment? Again, it will be helpful for the reader to have a basic understanding of the South African health care system. For example, in the United States, access barriers can arise from insurers’ unwillingness to pay for certain resources, and if a person cannot pay privately, they do without. Are the shortages in South Africa different in that resources are unavailable even if someone was wealthy enough to pay privately?

 Added a section on the South African health care in the introduction 

 Similarly, regarding prevention and care – what contributes to your reported non-compliance of health workers to various protocols – are they onerous to follow, lack of knowledge about their existence? What are the institutional influences that contribute? Added the reasons Discussed the reasons 302-305

In general, you could shorten the discussion somewhat by concisely addressing the issues specific to your findings and your overall goal of examining environmental barriers to SHC prevention in a lower-income nation. Revised 

Conclusion

• Conclusion: You introduce new material in the conclusion – Figure 1. This material would be better placed in the discussion, or perhaps even results section Removed the figure 

References

• Please correct reference #9 Revised

---

## [Decision Letter · Decision Letter 1]

15 Apr 2021

PONE-D-20-20175R1

“ Environmental factors influencing the prevention of secondary health conditions among people with spinal cord injury, South Africa.

PLOS ONE

Dear Dr. Pilusa,

Thank you for submitting your manuscript to PLOS ONE. After careful consideration, we feel that it has merit but does not fully meet PLOS ONE’s publication criteria as it currently stands. Therefore, we invite you to submit a revised version of the manuscript that addresses the points raised during the review process.

ACADEMIC EDITOR: Please see below for my comments on the manuscript.

We look forward to receiving your revised manuscript.

Kind regards,

Subas Neupane

Academic Editor

PLOS ONE

Journal Requirements:

Additional Editor Comments (if provided):

Please pay attention in the English language used. The English language must be checked by professional language editor. Few examples:

Line 52-54, the language should be revised, also the sentence in incomplete. “ We need to explore environmental factors influencing health outcomes, inform the context-based interventions for people with disabilities in South Africa, there.

In the methods part line 104 “We used a qualitative method was used to explore the environmental factors influencing the prevention of SHCs in people with SCI”.

Reviewers' comments:

Reviewer's Responses to Questions

**Comments to the Author**

1. If the authors have adequately addressed your comments raised in a previous round of review and you feel that this manuscript is now acceptable for publication, you may indicate that here to bypass the “Comments to the Author” section, enter your conflict of interest statement in the “Confidential to Editor” section, and submit your "Accept" recommendation.

Reviewer #1: All comments have been addressed

2. Is the manuscript technically sound, and do the data support the conclusions?

Reviewer #1: Yes

3. Has the statistical analysis been performed appropriately and rigorously? 

Reviewer #1: Yes

4. Have the authors made all data underlying the findings in their manuscript fully available?

Reviewer #1: Yes

5. Is the manuscript presented in an intelligible fashion and written in standard English?

Reviewer #1: Yes

6. Review Comments to the Author

Reviewer #1: Nice addition of South African health care system information in the introduction

Methods

Line 119 - OPD – outpatient department? Please define

Additional information about recruitment is sufficient

Good that interview guide was added

Sufficient additional detail to data analysis

Results

Better organized

Discussion

Restructured discussion section reads very well.

7. PLOS authors have the option to publish the peer review history of their article (what does this mean?). If published, this will include your full peer review and any attached files.

Reviewer #1: **Yes: **Anne M. Bryden

---

## [Author Response · Author response to Decision Letter 1]

27 Apr 2021

Corrections - highlighted in yellow in the manuscript

Editor 

 Corrected 

Citation (29)- removed this reference and replaced with a more recent article 

29. Maart S, Eide AH, Jelsma J, Loeb ME, Toni MK. Environmental barriers experienced by urban and rural disabled people in South Africa. Disabil Soc. 2007;22(4):357–69.

Replaced with: 

Hanass-Hancock J, Nene S, Deghaye N, Pillay S. ‘These are not luxuries, it is essential for access to life’: Disability related out-of-pocket costs as a driver of economic vulnerability in South Africa. African J Disabil [Internet]. 2017 [cited 2017 Nov 1];6(a280):1–10. Available from: https://doi.org/10.4102/ajod. v6i0.280%0A

Citation (8)

8. Mashola MK, Mothabeng J, Olorunju 

S. Readmission dues to secondary health 

conditions in people with spinal cord 

injury at a private rehabilitation facility in 

South Africa.Poster at WCPT congress 

2017, Cape Town. 2017;(21):2017. 

This reference is for a conference poster. 

Replaced the article with the article by 

Mashola et al. 2019 ( the same 

information from the same author)

Replaced with: 

Mashola MK, Olorunju SAS, Mothabeng J. Factors related to hospital readmissions in people with spinal cord injury in South Africa. South African Med J [Internet]. 2019 Jan 31;109(2):107. Available from: https://doi.org/10.7196/samj.2019.v109i2.13344

Line 268

Please pay attention in the English language used. The English language must be checked by professional language editor. Few examples:

Line 52-54, the language should be revised, also the sentence in incomplete. “ We need to explore environmental factors influencing health outcomes, inform the context-based interventions for people with disabilities in South Africa, there.

In the methods part line 104 “We used a qualitative method was used to explore the environmental factors influencing the prevention of SHCs in people with SCI”.

 Corrected grammar mistakes throughout the manuscript 

There is a need to explore environmental factors that influence health outcomes to inform context-based interventions for people with disabilities in South Africa.

We used a qualitative design to explore the environmental factors influencing the prevention of SHCs in people with SCI. 

Line 51-52

Line 103 

Reviewer 

Line 119 - OPD – outpatient department? Please define

 Changed to outpatient medical clinic Line 119

---

## [Editor Report · Decision Letter 2]

14 May 2021

“ Environmental factors influencing the prevention of secondary health conditions among people with spinal cord injury, South Africa.

PONE-D-20-20175R2

Dear Dr. Pilusa,

We’re pleased to inform you that your manuscript has been judged scientifically suitable for publication and will be formally accepted for publication once it meets all outstanding technical requirements.

Kind regards,

Subas Neupane

Guest Editor

PLOS ONE

Additional Editor Comments (optional):

Thank you for the revised manuscript. With this revision, the manuscript is potentially acceptable for publication in PLOS ONE.
---

## [Editor Report · Acceptance letter]

14 Jun 2021

PONE-D-20-20175R2 

Environmental factors influencing the prevention of secondary health conditions among people with spinal cord injury, South Africa. 

Dear Dr. Pilusa:

I'm pleased to inform you that your manuscript has been deemed suitable for publication in PLOS ONE. Congratulations! Your manuscript is now with our production department. 

Kind regards, 

on behalf of

Dr. Subas Neupane 

Guest Editor

PLOS ONE